# Identification of a Link between Suspected Metabolic Syndrome and Cognitive Impairment within Pharmaceutical Care in Adults over 75 Years of Age

**DOI:** 10.3390/healthcare11050718

**Published:** 2023-03-01

**Authors:** Zuzana Macekova, Tomas Fazekas, Michaela Krivosova, Jozef Dragasek, Viera Zufkova, Jan Klimas, Miroslava Snopkova

**Affiliations:** 1Department of Pharmacology and Toxicology, Faculty of Pharmacy, Comenius University in Bratislava, 832 32 Bratislava, Slovakia; 2Department of Physical Chemistry of Drugs, Faculty of Pharmacy, Comenius University in Bratislava, 832 32 Bratislava, Slovakia; 3Biomedical Centre Martin, Jessenius Faculty of Medicine in Martin, Comenius University in Bratislava, 036 01 Martin, Slovakia; 41st Department of Psychiatry, Faculty of Medicine, Pavol Jozef Šafárik University, 040 81 Košice, Slovakia; 5Department of Languages, Faculty of Pharmacy, Comenius University in Bratislava, 832 32 Bratislava, Slovakia; 6Department of Organisation and Management of Pharmacy, Faculty of Pharmacy, Comenius University in Bratislava, 832 32 Bratislava, Slovakia

**Keywords:** metabolic syndrome, older adults, community pharmacy, cognitive screening, cognitive impairment, pharmaceutical care

## Abstract

The prevalence of metabolic syndrome (MetS) and cognitive impairment (CI) is increasing with age. MetS reduces overall cognition, and CI predicts an increased risk of drug-related problems. We investigated the impact of suspected MetS (sMetS) on cognition in an aging population receiving pharmaceutical care in a different state of old age (60–74 vs. 75+ years). Presence or absence of sMetS (sMetS+ or sMetS−) was assessed according to criteria modified for the European population. The Montreal Cognitive Assessment (MoCA) score, being ≤24 points, was used to identify CI. We found a lower MoCA score (18.4 ± 6.0) and a higher rate of CI (85%) in the 75+ group when compared to younger old subjects (23.6 ± 4.3; 51%; *p* < 0.001). In the age group of 75+, a higher occurrence, of MoCA ≤ 24 points, was in sMetS+ (97%) as compared to sMetS− (80% *p* < 0.05). In the age group of 60–74 years, a MoCA score of ≤24 points was identified in 63% of sMetS+ when compared to 49% of sMetS− (NS). Conclusively, we found a higher prevalence of sMetS, the number of sMetS components and lower cognitive performance in subjects aged 75+. This age, the occurrence of sMetS and lower education can predict CI.

## 1. Introduction

The prevalence of both metabolic syndrome (MetS) and cognitive impairment (CI) is increasing with age [1,2]. According to the international classification of MetS [3] the prevalence of MetS ranged from 37% up to 60% in the elderly population [4,5,6]. Although cognitive impairments and dementia are often age-related disorders and according to World Health Organisation affect approximately 20–25% older population, they are not part of normal ageing [7]. In 2019 already over 55 million people worldwide suffer from cognitive disorders, AD, or dementia, and this number will almost double every 20 years, expect reaching 78 million in 2030 and 139 million in 2050 [7]. In general, MetS impairs overall intellectual functioning [1,2], and CI is the most significant factor of therapy failure in chronic disorders [8], mainly in older adults [9]. The presence of MetS, according to the classification of the International Diabetic Federation 2006 for the European population [3], can also be routinely evaluated in pharmaceutical care in a community pharmacy. For assessment of CI, Montreal Cognitive Assessment (MoCA) can be used as a simple, easy-to-use, but reliable cognitive screening tool [10,11] with high sensitivity for mild cognitive impairment [12]. 

Community pharmacists are the most accessible and frequently contacted healthcare professionals worldwide [13,14] who may play a crucial role in the identification of individuals with chronic disorders [15,16,17], including those suffering from cognitive disorders [18,19] in case that pharmacist is trained in the diagnosis of this type of disorder. Nowadays, common pharmaceutical care such as preparation, storage and dispensation of medicines, the provision of expert advice on their correct and safe use, or advice on the possibilities of non-pharmacological regimen measures is being globally expanded by other professional pharmacists’ competences (for example the monitoring of biochemical parameters, blood pressure measurement, management of obesity, smoking cessation, etc.) which are gradually becoming a part of pharmaceutical care worldwide which is more patient-oriented, defined as the expanded pharmaceutical care. Pharmaceutical care provided in nursing homes or senior care centres brings additional benefits to older adults [20,21]. Identification of potentially preventable risk factors (such as MetS and/or its components) and/or early stages of serious illnesses (e.g., cognitive impairment and dementia) within pharmaceutical care might help in slowing the rate of their progress and further disability [8,14,22]. Assessment of cognitive functions in elderly patients with MetS components is critical, but due to lack of time, it is routinely performed by only 24% of general practitioners, although 82% believe screening is needed [23]. Thus, the extension of pharmaceutical care toward cognitive screening might provide significant benefits for patients and the healthcare system.

The association between MetS and CI appears to be age-dependent [24,25]. The presence and onset of cardiovascular risk factors for CI are crucial for vascular modifications that result in reduced cerebral blood flow and metabolism in the brain [26]. While younger old (60–74 years) may be more susceptible to the cardiovascular load imposed by MetS on central neural pathways regulating mental processes [25], on the other side, MetS might have a positive influence on health status in older old (75+) individuals [27,28].

In our pilot study, we focused on the risk of suspected MetS (sMetS) estimated when providing healthcare service by a pharmacist and its related CI in the elderly [10,11] and showed the feasibility of cognitive testing in pharmaceutical care and its potential in identifying sMetS subject affected by CI but we did not investigate the impact of MetS in different age groups of elderly patients. We concluded that a quick and simple cognitive assessment could be a helpful extension of pharmaceutical care [10,11]. As our previous findings showed: (i) 56% of a random population over 60 years of age exhibited lower cognitive performance on the MoCA (ii) subnormal MoCA scores were significantly present with increasing age of the respondents, and (iii) the presence of MetS moderately but significantly correlated/associated negatively with the MoCA score [10]. Currently, in the same cohort as previously [10,11], we aimed to investigate whether sMetS has different effects on cognition in “younger old” (60–74 years) and “older old” (aged 75 years and over) individuals. Recent research reports diverse findings [1,22,24,29,30,31]. While MetS contributes to cognitive decline in “younger old” subjects [22,31], there is evidence that this effect may be weakened or vanished in 75+ individuals [24,29,30]. More detailed studies of the relationship between MetS and CI in the elderly population before the age of 75 and at the age of 75+ could have a global benefit [32], but further studies are needed. 

In this study, we aimed to investigate the impact of sMetS on cognition in aging individuals, with respect to the age category of 75+ years. Subjects were provided with pharmaceutical counselling, which means the specific patient-oriented pharmaceutical care service in community pharmacy targeted at the identification of components of MetS and MetS itself (according to IDF classification), including screening of cognitive features of enrolled older patients. We hypothesized that sMetS estimated within pharmaceutical care has a different influence on cognitive performance in a younger elderly population aged 60–74 years and in the 75+ population. We expected that younger old sMetS+ individuals will achieve significantly worse cognitive performance compared to the same age group without sMetS. On the other side, we expected that the cognitive performance in sMetS+ and sMetS− older old individuals will be either without difference or in the sMetS+ group only slightly weaker than in sMetS− group. 

## 2. Materials and Methods

### 2.1. Study Settings, Design and Sample Size

Here, we used data from a randomized pilot study in Slovakia [10,11], where 323 subjects were enrolled. Among them, 222 voluntary participants were interviewed in 16 community pharmacies, and 101 participants from 3 senior care centres aged 60 years and over were included, 63% in the 60–74 years group and 37% in the group 75+ (the age of the oldest participant was 95 years). 

### 2.2. Study Participants and Selection

The participants (68% women, 32% men), who visited a community pharmacy or lived in a senior care centre (between February 2018–February 2019) in Slovakia and who were willing to provide their general input data (socio-demographic information) and the list of all chronically used medications with the codes for their chronic diseases. Participants were randomly selected on the base of their voluntary consent and physical and mental ability to undergo screening. All respondents completed a simple data collection form in the Slovak language comprised of socio-demographic information (age, gender, education level), smoking and physical activity habits, and presence or absence of abdominal obesity, mediated by a pharmacist. The basic characteristics of the cohort sample are displayed in Table 1. Subsequently, a cognitive screening by the MoCA test was performed by trained pharmacists. Exclusion criteria were severe physical or mental health conditions that interfered with cognitive screening test realization and/or incompletely filled data collection form. We excluded 42 incompletely filled data collection forms. The forms were collected for one year (February 2018–February 2019), and the study was approved by the Ethics Committee of Faculty of the Pharmacy, Comenius University in Bratislava (EK FaF UK 01/2018). All procedures followed the relevant guidelines and regulations under the Declaration of Helsinki.

### 2.3. Classification of MetS and Assessment of Cognitive Function

According to provided codes for patients’ chronic diseases and information about the present/absence of abdominal obesity, there were identified individual components of MetS. Suspected metabolic syndrome (sMetS) was assessed according to the International Diabetes Federation Worldwide Definition of MetS, 2005, modified for the European population [3]. Accordingly, patients were divided with respect to the presence (sMetS+) or absence of suspected MetS (sMetS−).

The Montreal Cognitive Assessment is one of the available cognitive screening instruments, which scans seven cognitive domains: executive functioning; visuospatial abilities; language; attention, concentration and working memory; abstract reasoning; memory and orientation. The Slovak version of the Montreal Cognitive Assessment (MoCA) [8] with a reduced cut-off of ≤24 points for cognitive impairment by Bartos and Fayette was used [33] by pharmacists who were trained in the MoCA screening tool. Administration time was approximately 15 min, participant achieved a score between 0–30 points. 

### 2.4. Statistical Analysis 

Data were analysed using the SAS Education Analytical Suite for Microsoft Windows, version 9.3 (Copyright © 2012 SAS Institute Inc., Cary, NC, USA). The continuous demographic and clinical variables of study groups (e.g., age, the MoCA score) were represented by simple arithmetic mean, standard deviation, or 95% confidence interval. Categorical descriptive variables (e.g., sMetS status, MoCA status) were characterized by absolute frequencies and percentages. When comparing two groups with continuous data, a two-sample *t*-test was used. In addition, Pearson’s Chi-Square test and Fisher’s exact test of cross-tabulated data were performed to analyse the association between frequencies of categorical variables. The 0.05 significance level was used as a threshold for statistical significance for all tests, and 0.8 was taken as a minimally acceptable power of tests. 

Exogenous variables are independent of the error term (e.g., metabolic symptoms and cognitive function) and they may have a significant impact on the validity of the measurement. We investigated these terms by standard procedures of regression diagnostics and control procedures were applied, like sample randomization and matching, and finally the ANOVA method was used as a statistical control to reduce the possible effect of extraneous variables.

We used random allocation which is a technique that minimizes confounders and eliminates systematic bias by allocating individuals for treatment and control groups solely by a chance. We chose this method for its simplicity and effectiveness in eliminating distortion.

Due to the pilot nature of the study, we did not perform an exact a priori calculation of the number of participants according to the case-control methodology. However, the power of the performed tests was controlled by appropriate *post hoc* calculations.

We also suggested simple predictive analytics to forecast the impact of patients’ age, sMetS status, and education level on cognitive performance in the MoCA test. As exclusive predictors in this model, the age (dichotomic groups 60–74 years vs. 75+), sMetS status (sMetS+/sMetS−) or MetS components (central obesity, high blood pressure, dyslipidaemias, diabetes mellitus 2) and education level (dichotomic groups “lower education” for 12 years and less, vs. “higher education” for 13 years and more, were used. The calculated output data were the MoCA status (MoCA normal/MoCA lower cognitive performance). The success score of the prediction model was expressed by the evaluation of the confusion matrix in percentage.

## 3. Results

### 3.1. Prevalence of sMetS and Cognitive Impairment

The prevalence of sMetS in the study cohort was 18.5% in 60–74 years participants and 27% in 75+ (NS). On average, individuals 75+ achieved significantly lower MoCA score (18.4 ± 6.0) than patients aged 60–74 (23.6 ± 4.3). Lower cognitive performance (MoCA score ≤24) was more frequent in 75+ (85%) vs. participants aged 60–74 years (51%; *p* < 0.001). In both subcohorts (60–74 years vs. 75+), age had a significant influence on cognitive performance (*p* < 0.05; vs. *p* < 0.001, respectively).

### 3.2. Occurrence of sMetS and Patients’ Cognitive Performance 

sMetS influenced MoCA score in 75+ seniors (see Figure 1) as we found a significantly higher occurrence of lower cognitive performance in MoCA in 75+ with sMetS (97%), when compared to 75+ sMetS− group (80%; *p* < 0.05; r^2^ = 0.063), the difference was −1.99 points in MoCA mean (NS). In contrast, the MoCA score in younger seniors was unaffected by the presence of sMetS. In participants aged 60–74 years, the prevalence of lower cognitive performance according to MoCA was 63% in the sMetS+ group and 49% in sMetS− (NS; the difference was −1.21 points in MoCA mean, NS). 

### 3.3. Number of MetS Components and Patients’ Cognitive Performance

75+ individuals had a significantly higher number of MetS components (2.2 ± 0.9) than 60–74 participants (1.6 ± 1.1; *p* < 0.001) in both age groups, however, the number of MetS components was not associated with patients’ cognitive performance in MoCA.

### 3.4. Association between a MetS Status, Age, Education Level and Cognitive Performance

We proposed here a simple predictive model (see Figure 2) using three input categorical components, such as an occurrence or absence of sMetS and affiliation with a given age group (60–74 vs. over 75 years) and the observed output data expressed by the cognitive performance group (below or above the norm) with the success rate of classification of 73% (*p* < 0.001). The odds ratio for the age group 75+ against the youngers was 5.54; Cl 95% = 3.24–9.83 (*p* < 0.001), and this parameter for the occurrence of sMetS against the missing metabolic syndrome was as high as 2.04; CI 95% = 1.11–3.87 (*p* < 0.05), respectively. The odds ratio for the lower education group against the higher was 3.88; CI 95% = 1.87–8.46 (*p* < 0.001). We also performed an alternative predictive model based on the number of MetS components and patients’ cognitive performance expressed on the MoCA scale. The results of this model predicted a negative impact on the cognitive performance given by MoCA levels with the increasing number of MetS components (r = 0.44; *p* < 0.05) at the success rate of classification of 61%. The addition of other input parameters (gender, physical activity, smoking habits) that were available in the research did not improve the quality of the model.

## 4. Discussion

Previously, in a pilot study investigating the feasibility of cognitive screening within extended pharmaceutical care in elderly patients with sMetS [10,11], we reported that the population over 60 years of age exhibits lower cognitive performance in MoCA test and subnormal MoCA scores are significantly present with increasing age of study participants. In this investigation which widens previous findings, we hypothesized that sMetS has a different influence on cognitive performance in the younger elderly population aged 60–74 years and the 75+ population. The main results of the present study are as follows: (i) Presence of sMetS did not have a significant effect on achieved MoCA score in elderlies aged 60–74 years; (ii) sMetS has, thought moderate but significant, effect on achieved MoCA score in participants aged 75 years and more.

### 4.1. Prevalence of MetS and Cognitive Impairment in Elderly

Several recent studies reported that MetS increases the risk of developing CI or dementia for elderly patients aged 60–75 [22,31] but not in the 75+ elderly population [25,28,34]. These outcomes may have been related to a survival bias because participants with more severe MetS may have passed away earlier than reaching the older age [28]. Our findings did not show an association between sMetS and lower MoCA scores in participants aged 60–74 years compared to age-matched patients without sMetS. The potential explanations of controversy may lie in the possible influence of single MetS components as they strongly correlate with lower cognitive performance [2]. We can only speculate that there could be a more significant substantial influence of age than sMetS on CI in younger seniors.

### 4.2. Prevalence of MetS and Cognitive Impairment in Younger Elderly Patients

Recent studies conferred that MetS-related CI that has been observed in younger elderly participants aged 60–74 years [22,31] tends to diminish after reaching age 75+ [34] and can disappear or reverse in an oldest-old cohort [29,30]. Instead, our results showed the opposite, i.e., a minimal but significantly higher occurrence of MoCA ≤ 24 points in 75+ subcohort with sMetS when compared to the 75+ sMetS− group. Decelerated CI related to MetS was shown in the 75+ cohort [28], mainly in 85+ [30]. 

### 4.3. Prevalence of MetS and Cognitive Impairment in Older Elderly Patients

The presence of MetS in 75+ may be a protective evolutionary factor against the harmful aging process [28], and it may also have survival benefits in 75+ individuals with cardiovascular diseases [27,35]. Individuals with cardiovascular diseases who reached the age of 85+ may be relatively less susceptible to the adverse effects of MetS and its components [29,30]. Late-life MetS can also suppress the effects of other risk factors for the deterioration of cognitive features, such as malnutrition [36]. Weight loss may be a potential risk factor for CI or Alzheimer’s disease and a part of the process of dementia [37]. Our findings support the hypothesis that the effect of MetS on cognitive function with advancing age (after 75 years) is relatively weakened and that individuals with components of MetS aged 85+ years are probably more resistant to the effect of MetS on cognition. 

### 4.4. Coexistence of the Three Risk Factors: Occurrence of MetS, Age 75+, Lower Education Predicts Lower Cognitive Performance

Our predictive model for estimation of CI status was able to discriminate between individuals with (MoCA score ≤ 24) and without impaired cognitive functions (MoCA score >24) using three simple variables— the age group (60–74 vs. over 75 years, presence or absence of MetS and lower and higher education level) and this was superior to the predictive model using the number of MetS components. It might represent a simple tool for pharmacists to identify risk patients for CI who could need an individual approach in pharmaceutical care, e.g., control and management of modifiable risk factors for CI, revision of the medical list, and management of medication with potential risk for CI. Risk patients for CI also may undergo cognitive screening in a pharmacy and then be advised to visit a specialist when needed. Although previously suggested predictive models [38,39] reached higher predictive performance than ours, they used various parameters such as subjective well-being, educational level, marital status, and the presence of other chronic diseases obtained within the medical examination. The advantage of our predictive model lies in applying a few easy predictors to collect within routine pharmaceutical counselling.

### 4.5. Possible Pathological Background Explaining the Link between sMetS and CI

Previously [10], we reported an influence of the individual sMetS components, type 2 diabetes mellitus, hypertension and obesity, but not dyslipidaemias, on lower cognitive performance. This is also relevant to current findings.

First, numerous epidemical studies supported that diabetes is closely related to a higher risk of cognitive decline [40], including mild cognitive impairment and dementia. At the same time, cognitive dysfunction is increasingly recognised as an important comorbidity and complication of diabetes that affects patients’ quality of life, diabetes self-monitoring, and is related to diabetes treatment-related complications [41]. Watts and colleagues [42] reported that insulin is an important predictor of cognitive performance and decline, in opposite directions. In healthy older patients with normal cognition, higher insulin predicted greater cognitive impairment on attention and verbal memory. In contrast, in the group with early Alzheimer’s disease, higher insulin was associated with better cognitive performance in attention and verbal memory. In general, hyperglycaemia is associated with lower cognitive abilities and with a prevalence of mild cognitive impairment in elderly subjects [2] and achieved a score in test Mini-Mental State Examination is negatively correlated with fasting hyperglycaemia in the elderly population [2]. Diabetes is in close association with a high risk for hyperglycaemia and hypoglycaemia events, mainly in the elderly, which may be caused by the disease itself or by the glucose-lowering medication and may lead to impairment of cognitive features. Cognitive dysfunction can also predict these complications. Early identification of individuals, particularly in older age, with mild cognitive decline and adequate intervention, can improve adherence and may help to avoid later complications [41].

Second, a number of studies unveiled a relationship between high blood pressure and cognition in the elderly population. Their results showed a significant association between elevated blood pressure and lower cognitive performance in older subjects [2,43]. Combination of hypertension in midlife and low diastolic blood pressure in late-life were in relationship with reduction of brain volume and lower cognitive performance in the aging population [44,45]. In addition, longitudinal study demonstrated that long duration hypertension predicted cognitive decline independent of age [46]. In line with this, women at the age of 75 years had faster declines in global cognition associated with higher systolic blood pressure and lower diastolic blood pressure [47].

Third, also a relationship between obesity and worsened cognitive performance was investigated by many studies though outcomes are controversial. While being overweight is related to a lower risk for cognitive decline in the elderly population, central obesity increases the risk for it [48]. While obesity, as a component of MetS, in young and middle age means a risk factor for cardiovascular and cerebrovascular events [49], likewise weight loss later in life can mean an early warning signal for both development of Alzheimer’s disease and mild cognitive impairment [37]. The possible explanation may lay in a possible key link between obesity, but also other components of MetS, and cognitive decline as a consequence of inflammation and oxidative stress in the brain tissues [50].

### 4.6. Limitations

Our study has certain limitations in addition to cohort size. First, we used only the medication list of patients and diagnoses on the prescription to identify sMetS components. Second, pharmacotherapy of other possible morbidities was not analysed.

Also, possible biases might occur. The main sources of probable data distortion in our research are selection, information, and confounding bias. We assume the most significant contribution of selection bias. It is well known that age, education and estimated premorbid intelligence correlate significantly with the total MoCA score. Since it was a pilot study, the extent of these individual contributions was not estimated.

## 5. Conclusions

We found a higher prevalence of sMetS, the number of sMetS components and lower cognitive performance in MoCA in patients aged 75+. We confirmed the hypothesis that advancing age has a significant influence on cognition in both age groups (60–74 years vs. 75+). We observed a moderate but significant link between sMetS and CI exclusively in individuals aged 75+ but not in younger old participants. This finding confirms that metabolic syndrome substantially contributes to loss of cognitive performance during senescence, and it should also be considered when providing pharmaceutical services, particularly in adults aged 75+. Considering that forgetfulness or impaired memory is a common reason for low adherence in the elderly, early identification of elderly patients with potential cognitive impairment can help control modifiable risk factors for CI, prevent irregular medication use or non-adherence to medication and thus delay further complications. 

## Figures and Tables

**Figure 1 healthcare-11-00718-f001:**
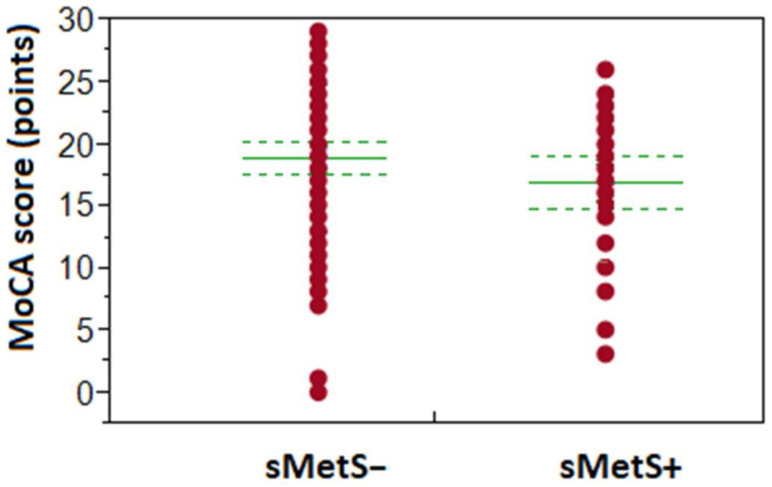
Impact of sMetS on the achieved MoCA score in the age group of 75+ years old participants. Group sMetS+ represents patient’s data with the presence of metabolic syndrome, and sMetS− are those without it. Solid lines in the centre of points (MoCA score) show the group means, while dotted lines represent boundaries of the appropriate confidence intervals.

**Figure 2 healthcare-11-00718-f002:**
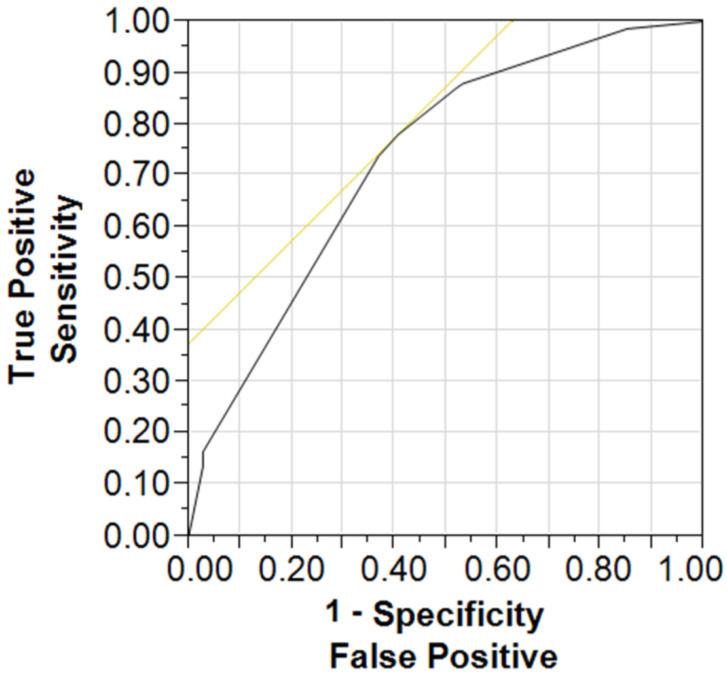
Receiver operating characteristic curve for a measure of discrimination of MoCA status (MoCA normal/MoCA lower cognitive performance) using three input variables (age, MetS status, education level).

**Table 1 healthcare-11-00718-t001:** Characteristics of respondents according to age groups.

Age Groups	60–74 Years N (%)	75+ Years N (%)
Participants, N (%)		
All	205 (63)	118 (37)
Gender, N (%)		
Female	128 (63)	91 (77)
Male	77 (37)	27 (23)
Age, median ± SD	67.1 ± 4.0	82.9 ± 4.1
Education (years), mean ± SD	12.3 ± 2.2	12.0 ± 2.4

## Data Availability

Data are available from the authors upon request.

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
