# Peer review of "Identification of a Link between Suspected Metabolic Syndrome and Cognitive Impairment within Pharmaceutical Care in Adults over 75 Years of Age"

_healthcare, 2023, doi:10.3390/healthcare11050718_

Round 1

Reviewer 1 Report

Dear Author

This is an interesting study to identify the link between suspected metabolic syndrome and cognitive impairment within pharmaceutical care in adults over 75 years of age. Overall, the manuscript was presented in a simple way, easily understandable for the readers. I have some queries:

- Line 89 - All respondents completed a simple questionnaire comprised of socio-demographic information (age, gender, education level), smoking and physical activity habits, and presence or absence of abdominal obesity.

Was the questionnaire in English? Was it self-administered? If so, was it self-administered even to the 95 year old participant?

- Line 93 - Exclusion criteria were severe physical or mental health conditions that interfered with cognitive screening test realization and/or incompletely filled questionnaires.

Was there any incompletely filled questionnaires? 

Line 95 - The questionnaires were collected for one year (February 2018 – February 2019), and the study was approved by the Ethics Committee of Faculty of Pharmacy, Comenius University in Bratislava (EK FaF UK 01/2018).

Almost there was 4 years lag in collecting the data and publication. Was it not a limitation of your study? Why was that lag? Does this lag make any difference in the results when compared to current elderly population? Is the ethical approval valid for more than 5 years?

Author Response

Reviewer 1

Dear Author

This is an interesting study to identify the link between suspected metabolic syndrome and cognitive impairment within pharmaceutical care in adults over 75 years of age. Overall, the manuscript was presented in a simple way, easily understandable for the readers.

We thank the reviewer for carefully reviewing the paper and for kind suggestions.

I have some queries:

#1: - Line 89 - All respondents completed a simple questionnaire comprised of socio-demographic information (age, gender, education level), smoking and physical activity habits, and presence or absence of abdominal obesity.

Was the questionnaire in English? Was it self-administered? If so, was it self-administered even to the 95 year old participant?

The data collection form was in Slovak language (due to the nationality of included patients, which was Slovak). It was administered by pharmacist who asked patients about their input data and wrote it on the data collection form. We extended the related section of Methods, as follows:All respondents completed a simple data collection form in Slovak language comprised of socio-demographic information (age, gender, education level), smoking and physical activity habits, and presence or absence of abdominal obesity, mediated by pharmacist.

#2: - Line 93 - Exclusion criteria were severe physical or mental health conditions that interfered with cognitive screening test realization and/or incompletely filled questionnaires.

Was there any incompletely filled questionnaires?

Yes, we had 42 incompletely filled data collection forms, which were excluded from study. We have revised the manuscript accordingly and added this information, as follows:

We excluded 42 incompletely filled data collection forms.

#3: Line 95 - The questionnaires were collected for one year (February 2018 – February 2019), and the study was approved by the Ethics Committee of Faculty of Pharmacy, Comenius University in Bratislava (EK FaF UK 01/2018).

Almost there was 4 years lag in collecting the data and publication. Was it not a limitation of your study? Why was that lag? Does this lag make any difference in the results when compared to current elderly population? Is the ethical approval valid for more than 5 years?

-We partially agree with reviewer, the 4 years delay could be considered a limitation of our study. However, after finishing the collection data, we immediately evaluated the impact of MetS and their components on cognition in our all-study population: our outcomes showed that MetS can accelerate a cognitive impairment present in elderly population. Pilot results were already published previously (1), as well as have been added in Introduction of revised manuscript. In current paper, as stated in Method, we continued in data analysis and we aimed to investigate the impact of sMetS on cognition in aging individuals, with respect to age category of 75+ years (i.e. aged 60-74 years vs. 75+ years).

-We are on the opinion that the lag in collecting the data and publication makes no difference in the results. There are still actual findings (2,3) suggesting that MetS and obesity is a significant issue and it is still considered as a high risk factor for cognitive decline.

-The approval of the ethics committee was not limited in time as well as the validity of the Ethical Consent was not limited in time.

  1. Macekova, Z.; Fazekas, T.; Stanko, P.; et al. Cognitive screening within advanced pharmaceutical care in elderly patients with suspected metabolic syndrome. Int J Gerontol. 2022, 16(4):355-360.
  2. Martocchia A, Gallucci M, Noale M, Maggi S, Cassol M, Stefanelli M, Postacchini D, Proietti A, Barbagallo M, Dominguez LJ, Ferri C, Desideri G, Toussan L, Pastore F, Falaschi GM, Paolisso G, Falaschi P; AGICO Investigators. The increased cortisol levels with preserved rhythmicity in aging and its relationship with dementia and metabolic syndrome. Aging Clin Exp Res. 2022 Nov;34(11):2733-2740.
  3. Pardo-Moreno T, Mohamed-Mohamed H, Rivas-Dominguez A, Garcia-Morales V, Garcia-Lara RA, Suleiman-Martos S, Bermudez-Pulgarin B, Ramos-Rodriguez JJ. Poor Cognitive Agility Conservation in Obese Aging People. Biomedicines. 2023 Jan 5;11(1):138. 

Reviewer 2 Report

Thank you for allowing me to review this manuscript. The subject is interesting, but great responsibility for cognitive diagnosis is also given to a group (pharmacists) who have no training in psychology. Therefore, there are parts of the manuscript that must be specified and improved.

1. The abstract should add a line of conclusions.

2. In the introduction, paragraph 1, statistical data on the proportion of increase with age of CI and MetS should be added. This would underline the importance of the investigation.

3. Cognitive disorders should ONLY be diagnosed by psychologists/psychiatrists and NEVER by pharmacists. Delete the last sentence of line 46 or specify "pharmacists trained in the diagnosis of this type of disorder".

4. In your hypothesis you talk about finding differences between the two age groups but you do not specify what difference is expected to be found, this must be clearly specified.

5. What do you mean when you talk about "undertaking pharmaceutical counselling"? You must specify it.

6. In the Participants section, specify the % of women and men even if it is in the Table.

7. For the questionnaire used, specify its general psychometric properties and calculate Cronbach's Alpha for your sample. This gives validity and reliability to the data found.

8. You say that the pharmacists passed the questionnaire, but were they trained or were they given some indications? This must be commented in the manuscript because a bad administration of the questionnaire can invalidate the results obtained.

9. Add in the table all the scores obtained from all the sociodemographic variables measured since you will later use them for the analyses.

10. The conclusions section is too brief, it should mark the importance of this research and future prospects. It must be expanded.

Author Response

Reviewer 2

Thank you for allowing me to review this manuscript. The subject is interesting, but great responsibility for cognitive diagnosis is also given to a group (pharmacists) who have no training in psychology. Therefore, there are parts of the manuscript that must be specified and improved.

We thank the reviewer for carefully reviewing the paper and for kind suggestions.

#1: The abstract should add a line of conclusions.

We thank the reviewer for this suggestion. Unfortunately, we are limited by journal instructions (200 words in abstract). We expanded conclusions in revised section of abstract, as follows:

Conclusively, we found a higher prevalence of MetS, the number of MetS components and lower cognitive performance in old subjects. This age, occurrence of MetS and lower education can predict CI.

#2: In the introduction, paragraph 1, statistical data on the proportion of increase with age of CI and MetS should be added. This would underline the importance of the investigation.

We added required statistical data to revised section of introduction, as follows:

According to international classification of MetS [3] the prevalence of MetS ranged from 37% up to 60% in elderly population [4, 5, 6]. Although cognitive impairments and dementia are often age-related disorders and according to World Health Organisation affect approximately 20–25% older population, they are not part of normal ageing [7]. In 2019 already over 55 million people worldwide suffer from cognitive disorders, AD or dementia and this number will almost double every 20 years, expect reaching 78 million in 2030 and 139 million in 2050 [7].

#3: Cognitive disorders should ONLY be diagnosed by psychologists/psychiatrists and NEVER by pharmacists. Delete the last sentence of line 46 or specify "pharmacists trained in the diagnosis of this type of disorder".

We completely agree with reviewer, the pharmacist could only identify individuals with potential risk of cognitive impairment and must be trained for this type of screening. We have specified this in manuscript as follows:

The Slovak version of the Montreal Cognitive Assessment (MoCA) [8] with a reduced cut-off of ≤24 points for cognitive impairment by Bartos and Fayette was used [33] by pharmacists who were trained in the MoCA screening tool.

#4:. In your hypothesis you talk about finding differences between the two age groups but you do not specify what difference is expected to be found, this must be clearly specified.

We thank the reviewer for this suggestion. We expanded this information in revised section of introduction, as follows:

We expected that younger old MetS+ individuals will achieve significantly worse cognitive performance compared to the same age group without MetS. On the other side, we expected that the cognitive performace in MetS+ and MetS- older old individuals will be either without difference or in MetS+ group only slightly weaker than in MetS- group.

#5: What do you mean when you talk about "undertaking pharmaceutical counselling"? You must specify it.

We apologize for imprecise wording. We revised it as as follows:

„receiving pharmaceutical care“.

#6: In the Participants section, specify the % of women and men even if it is in the Table.

We thank the reviewer for this suggestion. We added required information about participants in study in revised section of Methods as follows:

“(68% women, 32% men)”.

#7: For the questionnaire used, specify its general psychometric properties and calculate Cronbach's Alpha for your sample. This gives validity and reliability to the data found.

We thank the reviewer for this suggestion, and we apologize for imprecise wording. We revised Material and Methods as follows:

The Montreal Cognitive Assessment is one of available cognitive screening instruments, which scans seven cognitive domains: executive functioning; visuospatial abilities; language; attention, concentration and working memory; abstract reasoning; memory and orientation.

-Since we used the original version of the standardized data collection tool, in our case the calculation of the value of the Cronbach's alpha coefficient is not necessary, also due to the fact that a high value of the coefficient itself is not a guarantee of the unidimensionality of the research tool.

#8: You say that the pharmacists passed the questionnaire, but were they trained or were they given some indications? This must be commented in the manuscript because a bad administration of the questionnaire can invalidate the results obtained.

We confirm that pharmacists were trained to for data collection. We added this information in Material and Methods as follows:

“by pharmacists who were trained in the MoCA screening tool. Investigator responsible for a study (ZM) was trained for a neuropsychological and clinical diagnosis of cognitive disorders and to use a cognitive screening tools for a detection of mild cognitive impairment and Alzheimer disease (such as MoCA, etc.) in the Institute of Postgradual Medical Education in Prague (Czech Republic), and, subsequently, trained pharmacists for mediating the MoCA test. Pharmacists sent all completed tests to the responsible pharmacist for evaluation. Administration time was approximately 15 minutes, participant achieved score between 0 – 30 points.”

#9: Add in the table all the scores obtained from all the sociodemographic variables measured since you will later use them for the analyses.

The table already contains all the scores obtained from sociodemographic variables of participants such as  age (median ±SD), gender (F/M; %), education (years; mean ±SD). In addition, we have  collected information about smoking habits (yes/no), present of abdominal obesity (yes/no), regular physical activity (yes/no); medication list; the codes for chronic diseases. These variables were not expressed as score.

#10: The conclusions section is too brief, it should mark the importance of this research and future prospects. It must be expanded.

We thank the reviewer for this suggestion. We expanded the revised section of conclusion as follows:

We found a higher prevalence of MetS, the number of MetS components and lower cognitive performance in MoCA in patients aged 75+. We confirmed the hypothesis that advancing age has a significant influence on cognition in both age groups (60 – 74 years vs. 75+).

and

Considering that forgetfulness or impaired memory is a common reason for low adherence in elderly, early identification of elderly patients with potential cognitive impairment can help control modifiable risk factors for CI, prevent irregular medication use or non-adherence to medication and thus delaying further complication.

Reviewer 3 Report

I carefully reviewed this manuscript with a deep interest in this topic and the results are as follows.

1.       In theory, the higher the age, the higher the incidence of metabolic symptoms, which affects cognitive function damage, causing problems in self-drug management. This is a natural physical change due to human aging, but researchers have set it as a hypothesis, and research participants in group 2 (60 – 74 vs. 75+ years) confirmed the difference. Prior to conducting the study, it is well known that in common sense, the elderly population aged 75 or older is more likely to have problems with self-drug management than the elderly population aged 60-70 years. In this study, the research hypothesis should be established through academic grounds that can prove or disprove this common-sense fact.

2.      In this study, operational definitions of 'pharmacological care' and 'extended pharmacological care' are needed.

3.      In order to test the research hypothesis, exogenous variables such as metabolic symptoms and cognitive function must be controlled, and the method must be described.

4.      The random allocation method applied in this study should be specifically described.

5.      The basis for determining the number of samples of research participants should be described.

6.      Describe the location and duration of the study.

7.      Possible biases in this study should be described.

8.      Describe the dropout rate of partici The results of this study reported that there was a significant moderate relationship between sMetS and CI in the elderly population aged 70 or older. The meaning of this to the reader must be described in the discussion.pants in this study and why.

Author Response

Reviewer 3

I carefully reviewed this manuscript with a deep interest in this topic and the results are as follows.

We thank the reviewer for carefully reviewing the paper and for kind suggestions.

#1: In theory, the higher the age, the higher the incidence of metabolic symptoms, which affects cognitive function damage, causing problems in self-drug management. This is a natural physical change due to human aging, but researchers have set it as a hypothesis, and research participants in group 2 (60 – 74 vs. 75+ years) confirmed the difference. Prior to conducting the study, it is well known that in common sense, the elderly population aged 75 or older is more likely to have problems with self-drug management than the elderly population aged 60-70 years. In this study, the research hypothesis should be established through academic grounds that can prove or disprove this common-sense fact.

We thank the reviewer for this suggestion. We expanded the hypothesis in revised section of introduction as follows:

We expected that younger old sMetS+ individuals will achieve significantly worse cognitive performance compared to the same age group without sMetS. On the other side, we expected that the cognitive performace in sMetS+ and sMetS- older old individuals will be either without difference or in sMetS+ group only slightly weaker than in sMetS- group.

 #2: In this study, operational definitions of 'pharmacological care' and 'extended pharmacological care' are needed.

We added the explanations of „pharmaceutical care“ and „ expanded pharmaceutical care“ in revised section of introduction as follows:

Nowadays, a common pharmaceutical care such as a preparation, storage and dispensation of medicines, the provision of expert advice on their correct and safe use, or advice on the possibilities of non-pharmacological regimen measures is being globally expanded by other professional pharmacists´ competences (for example the monitoring of biochemical parameters, blood pressure measurement, management of obesity, smoking cessation, etc.) which are gradually becoming a part of pharmaceutical care worldwide which is more patient-oriented, defined as the expanded pharmaceutical care.

 #3: In order to test the research hypothesis, exogenous variables such as metabolic symptoms and cognitive function must be controlled, and the method must be described.

We thank the reviewer for this suggestion. We expanded these information in revised section of methods as follows:

Exogenous variables are independent of the error term (e.g. metabolic symptoms and cognitive function) and they may have significant impact on the validity of the measurement.  We investigated these terms by standard procedures of regression diagnostics and control procedures were applied like sample randomization and matching and finally the ANOVA method was used as a statistical control to reduce the possible effect of extraneous variables.

#4: The random allocation method applied in this study should be specifically described.

We thank the reviewer for this suggestion. We expanded the revised section of methods as follows:

We used random allocation which is a technique that minimizes confounders and eliminate systematic bias by allocation individuals for treatment and control groups solely by a chance.  We chose this method for its simplicity and effectiveness in eliminating distortion.

#5: The basis for determining the number of samples of research participants should be described.

We thank the reviewer for this suggestion.

Due to the pilot nature of the study, we did not perform an exact a priori calculation of the number of participants according to the case-control methodology. However, the power of the performed tests was controlled by appropriate post hoc calculations.

#6: Describe the location and duration of the study.

We thank the reviewer for this suggestion. We expanded the revised section of methods as follows:

“(between February 2018 – February 2019) in Slovakia .”

#7: Possible biases in this study should be described.

We thank the reviewer for this suggestion. We expanded the revised section of limitations as follows:

Also, possible biases might occur. The main sources of probable data distortion in our research are selection, information, and confounding bias. We assume the most significant contribution of selection bias. It is well known that age, education and estimated premorbid intelligence correlate significantly with the total MoCA score. Since it was a pilot study, the extent of these individual contributions was not estimated.

#8: Describe the dropout rate of partici The results of this study reported that there was a significant moderate relationship between sMetS and CI in the elderly population aged 70 or older. The meaning of this to the reader must be described in the discussion.pants in this study and why.

According to statistical analysis, we observed no dropout in the sample. According to exclusion criteria, we have excluded 42 incompletely filled data collection forms. We have revised the manuscript accordingly.

Also, we have extended the discussion according to reviewer´s suggestion, as follows:

Possible pathological mechanisms explaining the link between sMetS and CI

Previously [10], we reported an influence of the individual sMetS components:, type 2 diabetes mellitus, hypertension and obesity, but not dyslipidaemias, on lower cognitive performance. This is relevant also for current findings.

First, numerous epidemical studies supported that diabetes is closely related to higher risk of cognitive decline [40], including mild cognitive impairment and dementia. At the same time, cognitive dysfunction is increasingly recognised as an important comorbidity and complication of diabetes that affects patients´ quality of life, diabetes self-monitoring, and is related to diabetes treatment-related complications [41]. Watts and colleagues [42] reported that insulin is an important predictor of cognitive performance and decline, in opposite directions. In healthy older patients with normal cognition, higher insulin predicted greater cognitive impairment on attention and verbal memory. In contrast, in the group with the early Alzheimer disease, higher insulin was associated with better cognitive performance on attention and verbal memory. In general, hyperglycaemia is associated with lower cognitive abilities and with prevalence of mild cognitive impairment in elderly subjects [2] and achieved score in test Mini-Mental State Examination is negatively correlated with fasting hyperglycaemia in elderly population [2]. Diabetes is in close association with high risk for hyperglycaemia and hypoglycaemia events, mainly in the elderly, which may be caused by the disease itself or by the glucose-lowering medication and may lead to impairment of cognitive features. Cognitive dysfunction can also predict these complications. Early identification of individuals, particularly in older age, with mild cognitive decline and adequate intervention can improve adherence and may help to avoid later complications [41].

Second, number of studies unveiled a relationship between high blood pressure and cognition in elderly population. Theirs results showed significant association between elevated blood pressure and lower cognitive performance in older subjects [2, 43]. Combination of hypertension in midlife and low diastolic blood pressure in late-life were in relationship with reduction of brain volume and lower cognitive performance in aging population [44,45]. In addition, longitudinal study demonstrated that long duration hypertension predicted cognitive decline independent of age [46]. In line with this, women at age 75 years had faster declines in global cognition associated with higher systolic blood pressure and lower diastolic blood pressure [47].

Third, also a relationship between obesity and worsened cognitive performance was investigated by many studies though outcomes are controversial. While overweight is related to lower risk for cognitive decline in elderly population, central obesity increases risk for it  [48]. While obesity, as a component of MetS, in young and middle age means a risk factor for cardiovascular and cerebrovascular events [49], likewise weight loss later in life can mean an early warning signal for both development of Alzheimer´s disease and mild cognitive impairment [37]. The possible explanation may lay in a possible key link between obesity, but also other components of MetS, and cognitive decline as a consequence of inflammation and oxidative stress in the brain tissues [50].

  1. Sato, N.; Morishita, R. Roles of vascular and metabolic components in cognitive dysfunction of Alzheimer disease: short- and long-term modification by non-genetic risk factors. Front Aging Neurosci. 2013, 5:64.
  2. Biessels, G.J.; Whitmer, R.A. Cognitive dysfunction in diabetes: how to implement emerging guidelines. Diabetologia. 2020, 63(1):3-9.
  3. Watts, A.S.; Loskutova, N.; Burns, J.M.; Johnson, D.K. Metabolic syndrome and cognitive decline in early Alzheimer's disease and healthy older adults. J Alzheimers Dis. 2013, 35(2):253-265.
  4. Tsai, C.K.; Kao, T.W.; Lee, J.T.; et al. Increased risk of cognitive impairment in patients with components of metabolic syndrome. Medicine (Baltimore). 2016, 95(36):e4791.
  5. Shah, N.S.; Vidal, J.S.; Masaki, K.; et al. Midlife blood pressure, plasma β-amyloid, and the risk for Alzheimer disease: the Honolulu Asia Aging Study. Hypertension. 2012, 59(4):780-786.
  6. Muller, M.; Sigurdsson, S.; Kjartansson, O.; et al. Joint effect of mid- and late-life blood pressure on the brain: the AGES-Reykjavik study. Neurology. 2014, 82(24):2187-2195.
  7. Power, M.C.; Tchetgen, E.J.; Sparrow, D.; Schwartz, J.; Weisskopf, M.G. Blood pressure and cognition: factors that may account for their inconsistent association. Epidemiology. 2013, 24(6):886-893.
  8. Levine, D.A.; Galecki, A.T.; Langa, K.M.; et al. Blood Pressure and Cognitive Decline Over 8 Years in Middle-Aged and Older Black and White Americans. Hypertension. 2019, 73(2):310-318.
  9. Hou, Q.; Guan, Y.; Yu, W.; et al. Associations between obesity and cognitive impairment in the Chinese elderly: an observational study. Clin Interv Aging. 2019, 14:367-373.
  10. Yaneva-Sirakova, T.; Traykov, L.; Petrova, J.; Gruev, I.; Vassilev D. Screening for mild cognitive impairment in patients with cardiovascular risk factors. Neuropsychiatr Dis Treat. 2017, 13:2925-2934.
  11. Shalev, D.; Arbuckle, M.R. Metabolism and Memory: Obesity, Diabetes, and Dementia. Biol Psychiatry. 2017, 82(11):e81-e83.

Round 2

Reviewer 2 Report

All comments and suggestions have been taken into account and the necessary modifications have been made to the manuscript.

Reviewer 3 Report

The manuscript was properly revised according to the reviewer's comments. This is thought to be used as a basis for drug management for the elderly.